# Prompt-prompted Adaptive Structured Pruning for Efficient LLM Generation

**Harry Dong, Beidi Chen, Yuejie Chi**
Department of Electrical and Computer Engineering
Carnegie Mellon University
Pittsburgh, PA 15213, USA
{harryd,beidic,yuejiec}@andrew.cmu.edu

## Abstract

With the development of transformer-based large language models (LLMs), they have been applied to many fields due to their remarkable utility, but this comes at a considerable computational cost at deployment. Fortunately, some methods such as pruning or constructing a mixture of experts (MoE) aim at exploiting sparsity in transformer feedforward (FF) blocks to gain boosts in speed and reduction in memory requirements. However, these techniques can be very costly and inflexible in practice, as they often require training or are restricted to specific types of architectures. To address this, we introduce GRIFFIN, a novel *training-free* and *calibration-free* method that selects unique FF experts at the sequence level for efficient generation across *a plethora of LLMs with different non-ReLU activation functions*. This is possible due to a critical observation that many trained LLMs naturally produce highly structured FF activation patterns within a sequence, which we call *flocking*. Despite our method's simplicity, we show with 50% of the FF parameters, GRIFFIN maintains the original model's performance with little to no degradation on a variety of classification and generation tasks, all while improving latency (e.g. $1.29\times$ and $1.25\times$ speed-ups in Gemma 7B and Llama 2 13B, respectively, on an NVIDIA L40). Code is available at https://github.com/hdong920/GRIFFIN.

## 1 Introduction

Transformers (Vaswani et al., 2017) have demonstrated incredible capabilities across a plethora of domains (Lin et al., 2022; Khan et al., 2022; Nerella et al., 2023). Their large language model (LLM) successors (Touvron et al., 2023; Team et al., 2023; Jiang et al., 2023; 2024; Team et al., 2024; Anthropic, 2024) have pushed the bar higher, but these behemoths have become performative at the price of enormous amounts of computation and memory demands. One significant contributor is the model size itself. Not only is storage an issue, model layers tend to be wide and plenty, slowing down inference. Moreover, given the existence of sparse structures in LLMs, especially in feedforward (FF) blocks (Geva et al., 2020; Dettmers et al., 2022; Li et al., 2022; Liu et al., 2023), these models waste computation on intermediate features with little to no impact on the final result. For instance, it has been observed that in OPT-175B (Zhang et al., 2022), fewer than 5% of neurons in FF blocks have nonzero values per token (Liu et al., 2023), meaning 95% of the compute in each FF block is wasted. Usually consisting of around two-thirds of the parameters in an LLM, FF blocks can be serious memory and compute bottlenecks. These inefficiencies are highly problematic in latency-sensitive scenarios like in chatbots and autonomous vehicles.

There have been many methods to exploit sparsity in LLMs for efficiency gains, such as pruning model weights and constructing mixtures of experts (MoEs). Pruning removes low-impact pre-trained weights to reduce storage, yet this often does not translate to real speed-ups in practice, unless the pruning is done in a structured and hardware-friendly manner (Xia et al., 2022; Santacroce et al., 2023; Ma et al., 2023; Li et al., 2023; Xia et al., 2023) which typically causes greater performance deterioration. MoEs better preserve the

original performance by adaptively selecting subsets of the model to use per input but also come with drawbacks. Unless the model has been trained in this fashion (Fedus et al., 2022; Jiang et al., 2024), it will need to learn a cheap yet effective gating function (expert selection mechanism) and sometimes even require full fine tuning. Perhaps an even bigger weakness of many of these methods is the limited effectiveness and special considerations to strictly enforce ReLU-like activation functions (Zhang et al., 2021; Li et al., 2022; Csordás et al., 2023). In summary, pruning and MoEs provide enormous benefits but also come with steep challenges:

1. Structured sparsity is difficult and costly to enforce without serious performance degradation.
2. Adaptive selection of model subsets need to be cheap and accurate.
3. The method should be effective with various activation functions.

Daunting at first, these challenges become surmountable because of a simple observation of a phenomenon we call *flocking*, highly consistent sparse activations that persist throughout a sequence observed in many LLMs. *Flocking emerges in FF activations (inputs into the FF down projection) when we focus on a sequence's relative activation magnitudes instead of the absolute values.* Examples from Llama 2 7B and Gemma 7B are shown in Figure 1. *The key takeaway is that neurons which produce high relative magnitudes are naturally shared across tokens within a sequence*, as seen with the distinct vertical streaks. Increasingly bizarre, the two models have different architectures and different non-ReLU activation functions.

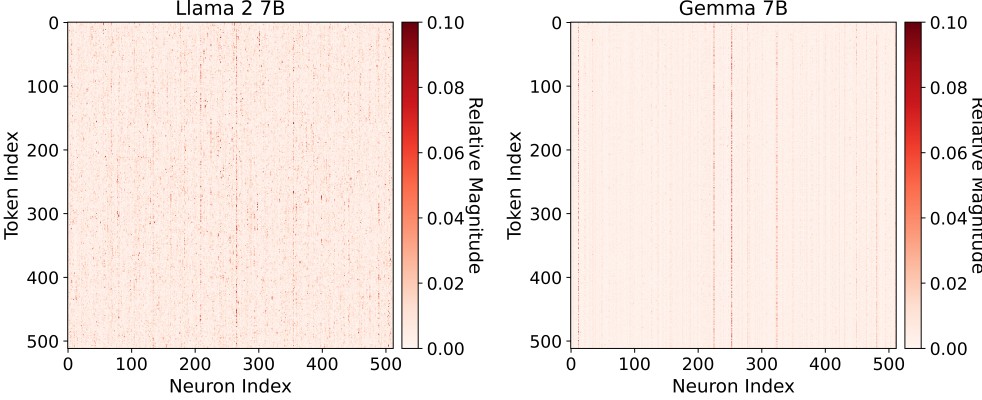

Figure 1: Relative FF activation magnitudes of the first 512 features and tokens across a sequence from PG-19 (Rae et al., 2019; Gao et al., 2020) in layer 10 of Llama 2 7B (left) and Gemma 7B (right). These heatmaps show flocking, where relative activation magnitudes are shared within a sequence, illustrated with the distinct dark vertical streaks. More examples in Appendix E.

Unlike existing pruning or MoE methods, we exploit flocking in our design of GRIFFIN (**G**ating by **R**epetition **I**n **F**eed**f**orward **I**ntermediate **N**eurons), *a highly performative and efficient training-free method to adaptively activate FF neurons.* GRIFFIN does this by using a sequence's prompt to determine the experts to activate during generation, allowing it to overcome all of the aforementioned challenges:

1. **No Preparation:** Our no-cost method is completely training-free and requires no calibration. Moreover, the simple implementation of GRIFFIN means it can be dropped into any FF block and instantly deployed.
2. **Simple Expert Selection:** Flocking in the prompt reveals the most relevant FF neurons for generation with little to no performance loss. The selection process is parameter-free and adds negligible overhead.
3. **Model & Activation Function Diversity:** Thorough experimentation demonstrates the efficacy of GRIFFIN on numerous models, including Llama 2 (Touvron et al.,

2023), Gemma (Team et al., 2024), Mistral (Jiang et al., 2023), OPT (Zhang et al., 2022), and ReluLlama (Team, 2023). Together, the tested activation functions include ReLU, SwiGLU, GEGLU, and ReGLU (Shazeer, 2020).

In this paper, we show GRIFFIN is a simple and strong adaptive structured pruning method because of flocking. In the next section (Section 2), we discuss in more detail some strengths and weaknesses of current methods that seek to improve FF efficiency. Then, we formalize the problem we are trying to solve, along with its motivation in Section 3. We present our novel approach in Section 4.2, which requires a thorough examination of the surprising phenomenon of flocking shared by many LLMs in Section 4.1. Our rigorous experiments demonstrate GRIFFIN preserves performance on classification and generation even after removing 50% of FF neurons (Section 5.1), all while having lower latency (Section 5.2). For instance, GRIFFIN reduces the number of active parameters in Llama 2 13B from 13B to 8.8B during generation to improve latency by $1.25\times$ with almost no loss in performance. Finally, we show our method's incredible scalability and robustness in several ablation studies (Section 5.3).

## 2 Background

Our novel method and FF activation observations are inspired and motivated by ample amounts of previous research that sought to characterize FF sparsity and accelerate LLMs.

**Feedforward Activation Sparsity.** The observation that transformer FF blocks produce sparse activations is not new (Geva et al., 2020; Dettmers et al., 2022; Li et al., 2022; Dong et al., 2023; Liu et al., 2023). In ReLU-based LLMs like OPT (Zhang et al., 2022), the activations can be exceptionally sparse and become more apparent for larger models (Liu et al., 2023). As more models use non-sparse activation functions like GLU variants (Shazeer, 2020), it is difficult for neurons to have no contribution to the output since these functions do not have an interval that maps to zero. Without exact sparsity, the efficacy of these methods becomes limited. As such, this has ushered a wave of models that are either adapted from available models (Zhang et al., 2021; Liu et al., 2021; Mirzadeh et al., 2023; Zheng et al., 2024; Jiang et al., 2024) or trained from scratch (Fedus et al., 2022) which can produce activations that are exactly zero with little to no performance loss. Even so, these methods require considerable amounts of computational resources.

**Pruning.** Pruning (LeCun et al., 1989) is one sparsity-guided way to tackle compute and memory bottlenecks of models. Previously, the common method would be some variation of iteratively rounding weights down to zero based on some score and retraining to recover any lost performance (Frankle & Carbin, 2018; Blalock et al., 2020; Liang et al., 2021; Liu et al., 2024). While this can result in most parameters being pruned, this method comes with a few issues. First, with the increasing scale of LLMs, retraining becomes impractical for most. Fortunately, cheap methods to effectively prune LLMs have been developed (Frantar & Alistarh, 2023; Sun et al., 2023; Jaiswal et al., 2023; Dery et al., 2024). The second issue is that unless pruning is done in a structured manner (Xia et al., 2022; Santacroce et al., 2023; Ma et al., 2023; Li et al., 2023; Xia et al., 2023), it is difficult to see real computational savings, yet structured pruning often leads to much more severe performance degradation. Third, pruning usually enforces sparsity to be static at inference which can be strong assumption since FF blocks are widely believed to contain the model's memory (Geva et al., 2020).

**Mixture of Experts.** Making sparsity more dynamic has motivated the design of mixture-of-experts (MoEs) (Jacobs et al., 1991) to avoid computing low-impact features in trained models at varying granularities (Zhang et al., 2021; Liu et al., 2023; Piórczyński et al., 2023; Csordás et al., 2023; Yerram et al., 2024; Zheng et al., 2024). The main idea of MoEs is to use a gating function to identify a small subset of neurons that will be used to compute the layer's output, an adaptive form of pruning. In the ideal case, all active neurons are selected and inactive neurons are ignored for each input. However, current methods either require training or rely on ReLU or ReLU-like activation functions.

## 3 Problem Formulation

This section contains an overview of different components of the FF block followed by a formulation of the FF compression problem which our method aims to tackle. Since FF blocks operate identically and independently for each token unlike attention, we begin with defining the FF block with a single column vector input $x \in \mathbb{R}^D$:

$$\text{FF}(x) = \text{FF}_2(\underbrace{\text{FF}_1(x)}_{z}) \tag{1}$$

where $\text{FF}_2(z) = W_2 z + b_2$ is a linear transformation and $\text{FF}_1$ is nonlinear. This describes a wide range of FF architectures and arbitrary activation functions $\sigma$. For instance, in OPT,

$$\text{FF}_1(x) = \sigma(W_1 x + b_1). \tag{2}$$

For FF blocks with GLU variants such as in Llama 2 and Gemma,

$$\text{FF}_1(x) = \sigma(W_g x + b_g) \odot (W_1 x + b_1) \tag{3}$$

where $\odot$ signifies element-wise multiplication. For all examples, $W_1, W_g \in \mathbb{R}^{D_{\text{FF}} \times D}$ and $W_2 \in \mathbb{R}^{D \times D_{\text{FF}}}$ where typically, $D_{\text{FF}} \gg D$. We refer to $z = \text{FF}_1(x)$ as the FF activations.

A popular method to compress FF blocks (which is also the goal of GRIFFIN, adaptive neuron pruning, and many MoE methods) is to find $\widehat{W}_1 \in \mathbb{R}^{k \times D}$, $\widehat{b}_1 \in \mathbb{R}^k$, and $\widehat{W}_2 \in \mathbb{R}^{D \times k}$ (additionally $\widehat{W}_g \in \mathbb{R}^{k \times D}$ and $\widehat{b}_g \in \mathbb{R}^k$ if needed) where $k < D_{\text{FF}}$ such that when the FF block is reparameterized with these matrices, the output value is preserved. Namely, for

$$\widehat{z} = \widehat{\text{FF}}_1(x) = \sigma(\widehat{W}_g x + \widehat{b}_g) \odot (\widehat{W}_1 x + \widehat{b}_1), \tag{4}$$

$$\widehat{\text{FF}}_2(\widehat{z}) = \widehat{W}_2 \widehat{z} + b_2, \tag{5}$$

$\text{FF}(x) \approx \widehat{\text{FF}}_2(\widehat{\text{FF}}_1(x))$, and similarly for FF blocks with non-GLU functions. For instance, in the MoE setting, these smaller matrices can vary from token to token and are actually selections of chunks of the original structures. Crucially, a solution to this problem leads to multiplication with smaller matrices which are naturally more efficient on GPUs and TPUs (Fatahalian et al., 2004; Wang et al., 2019). Note that for all equations defined in this section, they operate independently on each row of a length $S$ sequence input $X \in \mathbb{R}^{S \times D}$ (e.g. the activations for a sequence are $Z = \text{FF}_1(X) \in \mathbb{R}^{S \times D_{\text{FF}}}$).

## 4 From Flocking to GRIFFIN

Here, we take deeper dive into the phenomenon of flocking and describe the intuitive algorithm of GRIFFIN which is directly inspired by it.

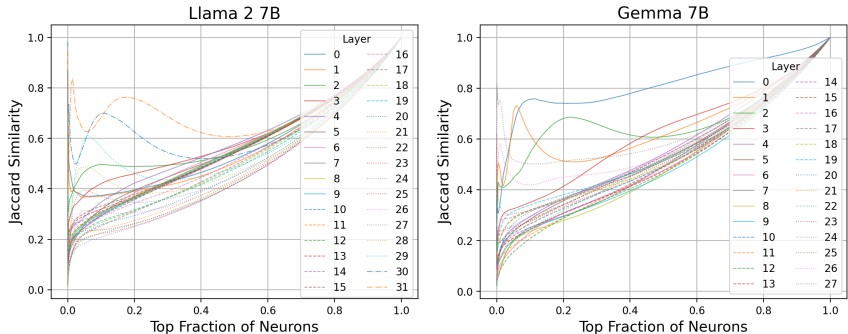

Figure 2: Average Jaccard similarity between WikiText samples' top FF neuron activations in Llama 2 7B (left) and Gemma 7B (right). Higher values indicate greater similarity.

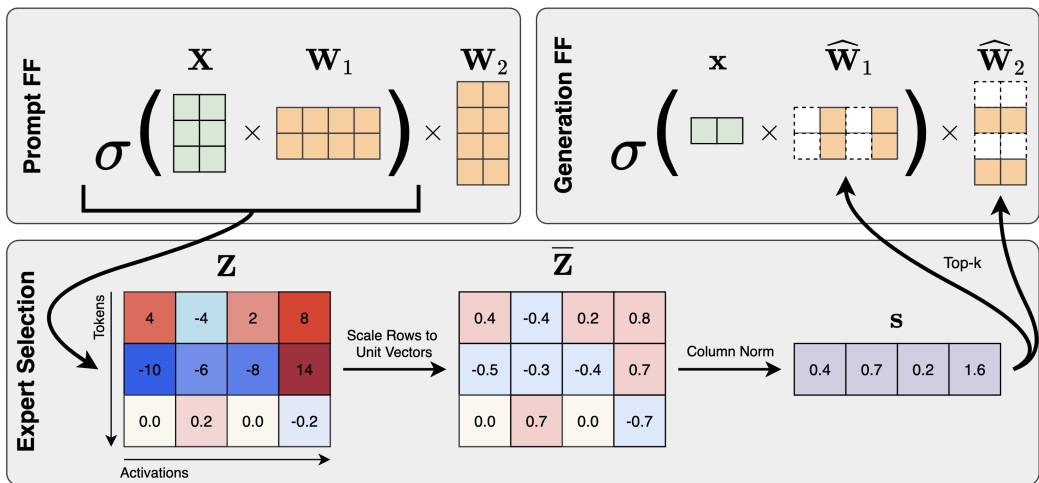

Figure 3: GRIFFIN overview. Relative activations from the prompt determine expert neurons to use for generation.

## 4.1 Observing Flocking

Flocking arises when we look at the relative impact of each neuron per token within a sequence. To see this, we normalize rows of $\mathbf{Z}$ to be unit vectors to construct $\overline{\mathbf{Z}} \in \mathbb{R}^{S \times D_{\text{FF}}}$ (i.e. $[\overline{\mathbf{Z}}]_i = [\mathbf{Z}]_i / \|[\mathbf{Z}]_i\|_2$), the *relative activations*. We show example relative activation magnitudes for a sequence in Llama 2 7B and Gemma 7B in Figure 1. Since there are distinct vertical streaks, this intriguingly implies that activations that have relatively greater weight are common across all tokens in a sequence. Notably, Llama 2 7B and Gemma 7B use SwiGLU and GEGLU activations, respectively, along with other major architecture differences. We call this phenomenon flocking, like highly organized groups of birds, and we observe this in virtually all FF layers (see Appendix E) and randomized inputs (see Appendix C).

While relative activations magnitudes are shared within a sequence, they are not generally shared between sequences. We show this by taking the $\ell_2$-norm of $\overline{\mathbf{Z}}$ along the token axis to obtain a length $D_{\text{FF}}$ vector for each sample or sequence, roughly capturing the contribution of each FF neuron throughout a sequence. Taking the top-$k$ of this for each sample at each layer, we find the Jaccard similarity between two sequences based on the indices selected for different $k$. In other words, we compute the intersection over union of every unique pair of top-$k$ sets. Higher values indicate more similar top-$k$ sets. From Figure 2 where we aggregate Jaccard similarities across WikiText (Merity et al., 2016) samples, we observe a lack of inter-sample activation similarities for the vast majority of layers in Llama 2 7B and Gemma 7B, unless the sets of selected neurons are large. *This lack of consistency implies statically pruning entire FF neurons without retraining would be less effective than a more adaptive method.*

## 4.2 GRIFFIN Algorithm

Using our insight on flocking, we introduce GRIFFIN as a simple general purpose and training-free adaptive structured pruning method for efficient generation, captured in Figure 3. In a nutshell, we select neurons during the prompt phase of each sample which are then used for the entire duration of the generation phase. This effective approach is based on a key observation on flocking: *since tokens within a sequence share activation patterns, the prompt and generated tokens will also share activation patterns.*

**Prompt Phase Expert Neuron Selection.** Our expert neurons are chosen at the sequence level, so we need to consider the dynamics of the entire input sequence rather than just a single token when choosing our neurons. To select expert neurons, we need a statistic

| MODEL | HELLASWAG | PIQA | COPA | ARC-E | ARC-C | BOOLQ |
|---|---|---|---|---|---|---|
| LLAMA 2 7B | 57.16 | 78.07 | 87.00 | 76.35 | 43.34 | 77.71 |
| MAGNITUDE | 57.12 | 77.31 | 84.00 | 70.33 | 40.27 | 66.54 |
| GRIFFIN | 57.11 | 77.69 | 86.00 | 74.54 | 42.75 | 73.15 |
| LLAMA 2 13B | 60.06 | 79.05 | 90.00 | 79.46 | 48.46 | 80.61 |
| MAGNITUDE | 60.00 | 79.00 | 88.00 | 74.07 | 46.25 | 70.52 |
| GRIFFIN | 60.10 | 79.11 | 89.00 | 77.19 | 46.84 | 78.50 |
| GEMMA 7B | 60.61 | 80.30 | 88.00 | 82.74 | 50.09 | 83.49 |
| MAGNITUDE | 46.24 | 73.12 | 57.00 | 45.20 | 32.76 | 62.84 |
| GRIFFIN | 60.62 | 79.98 | 88.00 | 81.65 | 50.09 | 81.90 |
| MISTRAL 7B | 61.21 | 80.58 | 92.00 | 80.89 | 50.43 | 83.61 |
| MAGNITUDE | 61.15 | 80.36 | 86.00 | 74.20 | 48.89 | 60.40 |
| GRIFFIN | 61.18 | 80.52 | 91.00 | 79.25 | 50.00 | 80.06 |
| OPT 6.7B | 50.48 | 76.28 | 81.00 | 65.53 | 30.55 | 66.12 |
| MAGNITUDE | 49.21 | 72.63 | 79.00 | 47.60 | 27.13 | 40.15 |
| GRIFFIN | 50.44 | 75.63 | 80.00 | 63.93 | 30.55 | 65.44 |
| OPT 13B | 52.46 | 75.90 | 86.00 | 67.05 | 32.94 | 65.81 |
| MAGNITUDE | 51.31 | 74.21 | 81.00 | 49.41 | 28.07 | 38.75 |
| GRIFFIN | 52.42 | 76.17 | 86.00 | 66.92 | 33.19 | 67.65 |

Table 1: Classification tasks (0-shot unnormalized accuracy) at 50% FF sparsity.

$s \in \mathbb{R}^{D_{\text{FF}}}$ to inform us of the importance of each neuron. At the prompt phase, we do this by taking the $\ell_2$-norm of $\overline{Z}$ along the token axis:

$$s = \left[ \|[\overline{Z}]_{\cdot,1}\|_2 \quad \cdots \quad \|[\overline{Z}]_{\cdot,D}\|_2 \right]^\top . \qquad (6)$$

Taking the indices of the top-$k$ across $s$ gives us the neurons we will use for this sample's generation phase which make up the set $\mathcal{E}$. Using the expert neurons in $\mathcal{E}$, we can find $\widehat{W}_1, \widehat{b}_1, \widehat{W}_g, \widehat{b}_g$, and $\widehat{W}_2$ by selecting corresponding rows and columns in $W_1, b_1, W_g, b_g$, and $W_2$, respectively. This is done for every FF block during the prompt phase. Illustrated in Appendix A, $s$ highlights neurons consistently activated at relatively high intensities.

**Generation with Expert Neurons.** When generating tokens, we directly use the pruned layers which contain the expert neurons, $\widehat{\text{FF}}_1$ and $\widehat{\text{FF}}_2$, to estimate $\text{FF}(X) \approx \widehat{\text{FF}}_2(\widehat{\text{FF}}_1(X))$ for all future tokens. In Llama 2 13B and Gemma 7B, this reduces the active number of parameters from 13B to 8.8B and from 8.5B to 5.4B, respectively, during generation.

## 5 Experiments

We showcase the superb performance of GRIFFIN on numerous tasks and models (Section 5.1) while achieving lower latency (Section 5.2), along with a study on several of its properties like sequence length scaling and batching (Section 5.3). All experiments are run on NVIDIA L40 GPUs.

### 5.1 Performance

Using various models, we evaluate on several generation and classification tasks. For generation, we evaluate on XSum (Narayan et al., 2018), CNN/DailyMail (Nallapati et al., 2016), COQA (Reddy et al., 2019), and SCROLLS QASPER (Dasigi et al., 2021; Shaham et al., 2022). For classification, we evaluate on HellaSwag (Zellers et al., 2019), PIQA (Bisk et al., 2020), COPA (Roemmele et al., 2011), ARC-Easy/Challenge (Clark et al., 2018), and BoolQ (Clark et al., 2019). With the exception of XSum and CNN/DailyMail, we use LM Evaluation Harness for our experiments (Gao et al., 2023). Aside from comparing with the original LLM, we also compare GRIFFIN with a static neuron pruning method based on neuron magnitudes. Similar to neuron magnitude pruning, this baseline selects expert

| MODEL | XSUM (ROUGE-1/2/L) | CNN/DAILYMAIL (ROUGE-1/2/L) | COQA (F1/EM) | QASPER (F1) |
|---|---|---|---|---|
| LLAMA 2 7B | 27.15/9.06/22.62 | 10.08/0.13/9.55 | 77.35/63.88 | 26.31 |
| MAGNITUDE | 9.71/1.31/8.59 | 9.66/0.63/9.32 | 56.59/39.93 | 12.93 |
| ADAPTIVE WANDA | 25.59/8.18/21.34 | 9.90/0.26/9.39 | 77.16/63.53 | 27.61 |
| GRIFFIN | 24.75/7.41/20.55 | 10.97/0.66/10.37 | 77.18/63.58 | 25.76 |
| LLAMA 2 13B | 26.90/9.45/22.09 | 2.51/0.22/2.34 | 79.18/66.37 | 28.32 |
| MAGNITUDE | 5.72/0.78/5.06 | 0.02/0.00/0.02 | 65.69/47.87 | 15.55 |
| ADAPTIVE WANDA | 24.65/8.08/20.29 | 1.13/0.11/1.07 | 79.43/67.43 | 27.98 |
| GRIFFIN | 25.69/7.85/20.89 | 3.31/0.78/3.07 | 79.22/66.62 | 27.91 |
| GEMMA 7B | 26.86/9.15/22.03 | 17.45/4.15/15.94 | 79.04/65.25 | 30.78 |
| MAGNITUDE | 1.49/0.01/1.47 | 0.00/0.00/0.00 | 2.92/1.50 | 7.02 |
| ADAPTIVE WANDA | 24.76/7.79/20.39 | 12.10/2.21/11.20 | 75.79/61.87 | 30.62 |
| GRIFFIN | 25.86/7.81/20.93 | 18.26/4.75/16.58 | 78.52/64.62 | 27.37 |
| MISTRAL 7B | 28.67/10.21/23.64 | 0.28/0.01/0.28 | 80.70/67.30 | 24.56 |
| MAGNITUDE | 3.58/0.27/3.31 | 0.26/0.03/0.26 | 61.99/45.93 | 17.18 |
| ADAPTIVE WANDA | 25.42/8.40/21.16 | 0.16/0.00/0.14 | 80.68/67.60 | 24.73 |
| GRIFFIN | 26.59/8.70/22.17 | 1.26/0.21/1.17 | 80.15/66.50 | 23.92 |
| OPT 6.7B | 23.60/7.04/19.46 | 13.85/1.54/13.04 | 68.70/54.98 | 18.53 |
| MAGNITUDE | 1.63/0.00/1.54 | 1.20/0.00/1.17 | 31.53/16.52 | 7.28 |
| ADAPTIVE WANDA | 22.40/6.04/18.57 | 12.94/1.14/12.23 | 69.01/55.22 | 18.47 |
| GRIFFIN | 21.17/5.42/17.58 | 13.01/1.06/12.26 | 68.99/55.00 | 17.40 |
| OPT 13B | 25.14/7.93/20.80 | 13.22/1.18/12.46 | 69.51/55.67 | 20.58 |
| MAGNITUDE | 1.23/0.00/1.21 | 1.29/0.00/1.29 | 39.38/27.07 | 8.87 |
| ADAPTIVE WANDA | 23.68/6.97/19.71 | 13.58/1.48/12.82 | 69.60/56.03 | 21.30 |
| GRIFFIN | 22.11/6.28/18.29 | 12.92/1.13/12.20 | 69.07/54.83 | 20.16 |
| RELULLAMA 2 7B | 25.10/7.81/20.76 | 20.95/6.79/19.24 | 78.49/66.73 | 23.31 |
| MAGNITUDE | 9.09/0.22/8.20 | 8.50/0.14/8.17 | 19.43/6.48 | 7.21 |
| ADAPTIVE WANDA | 22.62/6.47/18.82 | 17.46/5.56/16.13 | 78.96/66.97 | 21.38 |
| GRIFFIN | 21.83/5.88/18.09 | 16.85/4.96/14.69 | 78.35/67.10 | 22.29 |

Table 2: Generation tasks XSum (1-shot), CNN/DailyMail (1-shot), CoQA (0-shot), SCROLLS QASPER (0-shot) at 50% FF sparsity. Magnitude neuron pruning fails in almost every case while GRIFFIN effectively preserves performance.

neurons based on neuron magnitudes in $W_1$ for the generation phase but uses the entire FF blocks for prompting like GRIFFIN. In the case of GLU variants, the neuron-wise norms of $W_1$ and $W_g$ are elementwise multiplied to produce the pruning metric. As we will see, this straightforward baseline achieves great classification results but falters for generation. Another baseline we consider also uses the full model for the prompt and Wanda (Sun et al., 2023) in FF blocks for generation, using the prompt activations to prune FF weights. We refer to this adaptive *unstructured* pruning baseline as Adaptive Wanda. Note that Adaptive Wanda is completely unstructured and does not reduce the FF activation dimension at all.

As our method is designed specifically for generation, we alter classification evaluations to simulate generation. In typical classification tasks, LLMs do not enter the generative phase since the final token output of the prompt phase indicates the class. Consequently, directly applying GRIFFIN for classification tasks trivially yields the exact performance of the original model. Therefore, we treat all tokens but the final input token as the prompt. Then, the model is forced to go into the generation phase for one step.

We start with a look into the relationship between the sparsity levels and performance degradation. This translates to varying $k$ when we select the top-$k$ of our statistic $s$. To compare the performance degradation across multiple tasks, we plot the ratio of the final performance metrics between GRIFFIN and the full model in Figure 4. We see most of the performance is preserved at 50% FF sparsity in Llama 2 7B, Gemma 7B, and Mistral 7B. Different tasks have different tipping points where performance sharply drops, which may be related to the difficulty of the task (Yin et al., 2024).

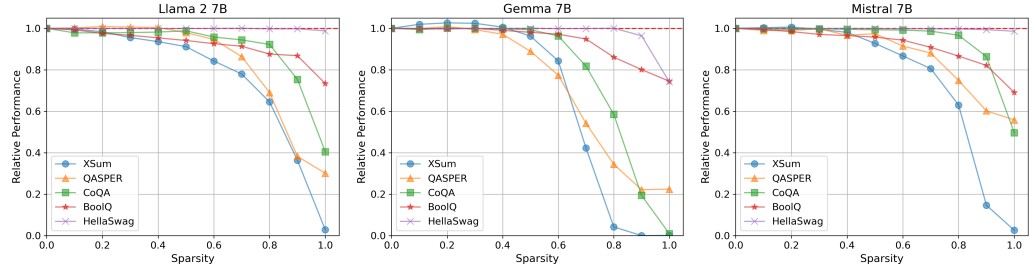

Figure 4: Relative performance of GRIFFIN for Llama 2 7B (left), Gemma 7B (center), and Mistral 7B (right) as we enforce varying degrees of sparsity per FF block. For all tasks, the original model's performance for each task is normalized to 1.

| MODEL | SETUP | PROMPT | FULL | MAGNITUDE | GRIFFIN |
|-------|-------|--------|------|-----------|---------|
| LLAMA 2 13B | 2048+128 | 0.5 | 6.8 | 5.4 / 5.0 | 5.4 / 5.1 |
| | 2048+2048 | 0.5 | 119.1 | 95.0 / 83.4 | 94.9 / 82.8 |
| GEMMA 7B | 2048+128 | 0.3 | 4.5 | 4.1 / 4.2 | 4.2 / 4.1 |
| | 2048+2048 | 0.3 | 87.1 | 67.7 / 65.0 | 67.4 / 65.0 |

Table 3: Generation phase latency (s). We denote "$P + G$" as the task of generating $G$ tokens from a length $P$ prompt. When relevant, times are in the format 50% / 75% FF sparsity.

Fixing FF sparsity to be 50%, we evaluate on more tasks and models. Table 1 and Table 2 show the performance of GRIFFIN on classification and generation, respectively. We see that magnitude neuron pruning achieves reasonable results for classification but completely annihilates the original performance in most generation settings. For generation, Adaptive Wanda and GRIFFIN perform similarly, preserving most of the performance for all models. However, GRIFFIN is able to accomplish this in a structured manner which allows it to be more efficient in practice. For example outputs, see Appendix D. Base models Mistral 7B and Llama 2 13B perform poorly on 1-shot CNN/DailyMail, as they often repeated "\n" or items from the example shot. With 3 shots, both obtain Rouge-1 scores above 25.

## 5.2 Efficiency

We now present efficiency metrics of GRIFFIN. We collect synthetic datasets with samples having identical lengths and average results across samples. Like many MoE methods, GRIFFIN is ideal for single sample inputs since it is adaptive, such as in the case of personal devices, so we use batch size 1 for these experiments. Using Hugging Face implementations of Llama 2 13B and Gemma 7B at FP16 precision, we measure the latency in different scenarios on an NVIDIA L40 GPU.

Recalling that our magnitude selection baseline is essentially neuron pruning at generation, this has the best possible speed-up since there is no expert neuron selection overhead per sample. From Table 3, GRIFFIN matches the best case, producing up to a 1.29× and 1.25× improvement in latency for long generation at 50% FF sparsity in Gemma 7B and Llama 2 13B, respectively. This illustrates that our method is as fast as a static neuron pruned LLM during generation while being adaptive to preserve the accuracy of the full model. In offloading settings with large models, our method has the potential to further accelerate inference. For a prompt, GRIFFIN essentially performs structured pruning on the massive network, and if this pruned model can fit on a single device, it will avoid offloading for the entirety of generation.

## 5.3 Ablations and Analysis

**Prompt vs. Generation Length.** We find that GRIFFIN can potentially be made more robust for long generation by lengthening the prompt. To see this, we use language modeling

on the concatenated version of WikiText to simulate generation. For a length $S$ input into the FF block, we designate the first $P$ tokens as the prompt and the last $G$ tokens as the generated portion such that $P + G = S$. The prompt partition is used to calculate our statistic $s$ and determine the expert neurons. The prompt partition uses the full FF block while the generation partition only uses the selected neurons. When comparing the original model with GRIFFIN, we only compute the perplexity of the outputs from the generation partition since the other outputs will be identical. Based on Figure 5, GRIFFIN gets closer to the full model outputs when the prompt length increases and generation length decreases, meaning the difficulty with long generation can be suppressed with longer prompts.

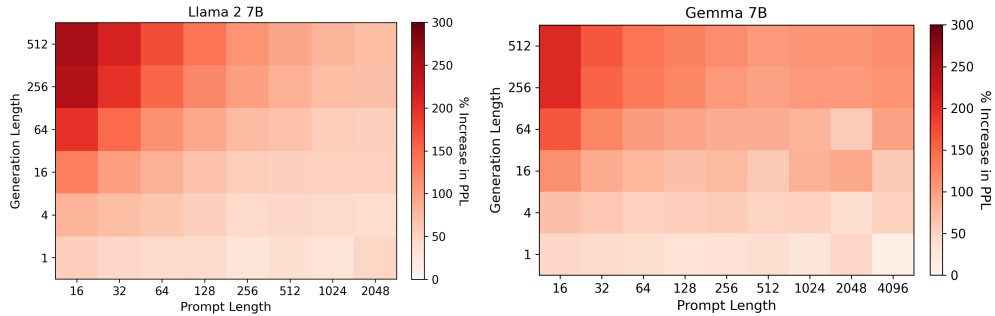

Figure 5: Prompt length vs. generation length for Llama 2 7B (left) and Gemma 7B (right) as measured by increase in perplexity (PPL) from the full model on concatenated WikiText at 50% FF sparsity.

**Sharing Selected FF Neurons.** We further verify that GRIFFIN is simultaneously robust to batching and more performative than fixed neuron pruning methods using similar pruning metrics to $s$ in (6). As such, GRIFFIN maintains its reliability as we extend beyond single inputs.

For static pruning, we test two methods. One is to use the example shot in the prompt to compute $s$. Then, for all samples with the same shot, we select the same FF neurons for generation. With the other method, we can extend (6) such that $\bar{s}$ can be a pruning metric for multiple inputs (i.e. the same FF neurons will be pruned). Suppose that $s_i$ is the metric for sample $i$ with prompt length $S_i$. Then, we use

$$\bar{s} = \sum_i \frac{s_i}{\sqrt{S_i}} \tag{7}$$

to select FF neurons. Finding a global $\bar{s}$ across all prompts in a dataset is the second method. Therefore, for a particular dataset, both methods have fixed expert neurons across samples.

To extend GRIFFIN to larger batch sizes, we use $\bar{s}$ from (7) where the samples in a batch are aggregated. Thus, adaptability is maintained. From Table 4, GRIFFIN (even with batch size >1) achieves better performance than the static methods, reinforcing the performance benefit of adaptability. Interestingly, even though performance decays as we increase the batch size, the decay is slow.

| MODEL | FULL | SHOT | GLOBAL | GRIFFIN (1) | GRIFFIN (4) | GRIFFIN (16) |
|---|---|---|---|---|---|---|
| LLAMA 2 7B | 27.15 | 21.11 | 23.28 | 24.75 | 24.26 | 23.75 |
| GEMMA 7B | 26.86 | 19.76 | 22.48 | 25.86 | 25.01 | 24.37 |

Table 4: Rouge-1 scores of 1-shot XSum with varying ways to use the same selected neurons across multiple samples. From left to right, the scores are the result of the full model, FF neurons selected based on the prompt, FF neurons selected based on the entire dataset, and GRIFFIN in batch sizes of 1, 4, and 16. All pruning methods use the full model for the prompt and 50% of the FF neurons during generation.

## 6 Conclusion

In this work, we have shown a special form of sparsity in FF layers and a simple method to exploit it. Flocking is a curious phenomenon present in many LLMs where tokens within a sequence activate neurons at similar intensities. This structure motivated the design of GRIFFIN, a learning-free adaptive structured pruning mechanism to remove FF neurons during inference at the sequence level which preserves the full model's performance on a large collection of classification and generative tasks at 50% FF sparsity while achieving lower latency. Furthermore, its applicability extends beyond just ReLU-based LLMs. With its straightforward algorithm and no-cost deployment, GRIFFIN expands the accessibility of numerous LLMs for generative inference.

#### Acknowledgments

We thank Zixin Wen for insightful discussions. The work of H. Dong is supported in part by the Wei Shen and Xuehong Zhang Presidential Fellowship at Carnegie Mellon University. The work of Y. Chi is supported in part by the grants NSF DMS-2134080 and ONR N00014-19-1-2404.

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

## A    Pruning Metric

Here we present visualizations of our gating statistic $s$ from (6). For a single sample, we find $s$ and sort the entries normalized between 0 and 1 in Figure 6. In both models, values in $s$ are heavily concentrated in a handful of features. Since $s$ aggregates the relative activation magnitudes across tokens, this implies $s$ can capture heavily and frequently activated neurons.

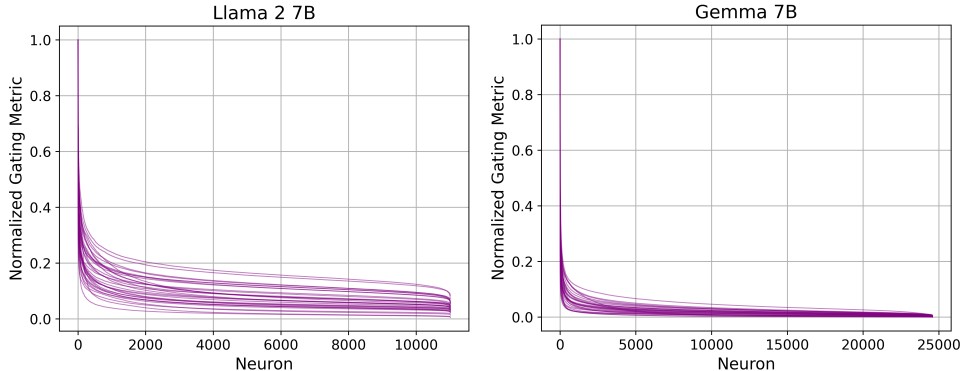

Figure 6: Sorted entries of $s$ for each layer of Llama 2 7B (left) and Gemma 7B (right) with a sequence from PG-19 as the input. Each line is a layer.

## B    Sampling-based Selection

Here, we verify that given the statistic $s$, top-$k$ expert selection produces better results than sampling-based methods. The methods we compare against include sampling based on the weights in $s$ and combining top-$k$ selection for half of the experts followed by weighted sampling. Based on Table 5, we can see that sampling generally degrades performance much more.

Table 5: Comparison between different expert selection methods at 50% FF sparsity.

| SELECTION METHOD | XSUM (ROUGE-1/2/L) | CNN/DAILYMAIL (ROUGE-1/2/L) | COQA (F1/EM) | QASPER (F1) |
|---|---|---|---|---|
| *Llama 2 7B* | | | | |
| FULL | 27.15/9.06/22.62 | 10.08/0.13/9.55 | 77.35/63.88 | 26.31 |
| TOP-$k$ | **24.75/7.41/20.55** | **10.97/0.66/10.37** | **77.18**/63.58 | **25.76** |
| SAMPLING | 21.04/5.22/17.12 | 8.78/0.49/8.28 | 76.15/62.53 | 24.46 |
| TOP-$k$ + SAMPLING | 24.35/7.08/20.07 | 10.45/0.48/9.88 | 77.12/**64.17** | 25.22 |
| *Gemma 7B* | | | | |
| FULL | 26.86/9.15/22.03 | 17.45/4.15/15.94 | 79.04/65.25 | 30.78 |
| TOP-$k$ | **25.86/7.81/20.93** | **18.26/4.75/16.58** | **78.52/64.62** | **27.37** |
| SAMPLING | 20.25/5.16/16.79 | 8.34/1.71/7.72 | 75.02/59.93 | 24.97 |
| TOP-$k$ + SAMPLING | 24.47/7.43/19.98 | 10.93/2.60/9.98 | 76.76/62.12 | 27.09 |

## C    Sparsity in Random Sequences

As further exploration into flocking, we investigate this phenomenon with random inputs. As input sequences, we use a sample from concatenated WikiText, a permuted version of that sample, and completely random sequence where tokens are uniformly sampled from the vocabulary. Seen in Figure 7, this structure exists in permuted and random inputs,

perhaps even more consistently than in unperturbed sequences. This suggests something within language actually diversifies the activations, the cause of which would be of interest for future work.

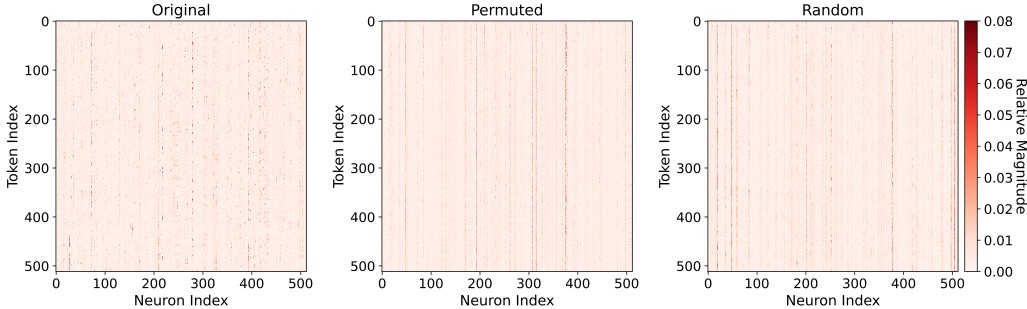

Figure 7: First 512 tokens and their relative FF activation magnitudes in layer 18 of Gemma 7B when inputting the original WikiText sequence (left), permuted sequence (center), and random tokens (right). Best viewed zoomed in.

# D  Generation Examples

We show example generated text in Figure 8. Adaptive Wanda and GRIFFIN produce similar summaries compared to the those produced by the full models and the target.

# E  More Flocking Examples

We provide more example of flocking across different layers of the LLM. Figure 9 and Figure 10 show flocking in Gemma 7B. Figure 11 and Figure 12 show flocking in Llama 2 7B. Flocking in Gemma 7B is more visually distinct while activations in Llama 2 7B are more distributed.

**Prompt**

<shot 1>

###
Article: The systems, at Kentucky Methodist Hospital, Chino Valley Medical Center and Desert Valley Hospital, California, are now running normally again.
None of the hospitals is believed to have paid the ransom.
And the cases are now being investigated by the FBI.
The Kentucky Methodist Hospital had to shut down all of its desktop computers and activate a back-up system.
A message on its homepage said: "Methodist Hospital is currently working in an internal state of emergency due to a computer virus that has limited our use of electronic web-based services.
"We are currently working to resolve this issue, until then we will have limited access to web-based services and electronic communications."
It later said no patient data or care had been affected.
Fred Ortega, a spokesman for Prime Healthcare Services, which owns Chino Valley Medical Center and Desert Valley Hospital, said: "It did cause significant disruptions of our IT systems.
"However, most of the systems and the critical infrastructure has been brought back online."
The attack comes weeks after it was revealed Hollywood Presbyterian Medical Centre in Los Angeles had been attacked by ransomware.
In that case, it paid $17,000 to get access to files back.
Kentucky Methodist Hospital information systems director Jamie Reid named the malware involved as Locky, a new bug that encrypts files, documents and images and renames them with the extension .locky.
The most common way Locky gets itself on machines is via a spam email with an attached document that looks like nonsense and advises readers to enable macros "if the data encoding is incorrect".
Once the malware is downloaded, it sends a message to desktops with instructions about how users can pay to have files unlocked.
In November, a report from Intel's McAfee labs said the number of ransomware attacks was expected to grow in 2016.
Security expert Brian Krebs said: "It's a fair bet that as ransomware attacks and attackers mature, these schemes will slowly become more targeted.
"I also worry that these more deliberate attackers will take a bit more time to discern how much the data they've encrypted is really worth, and precisely how much the victim might be willing to pay to get it back."

Summarize the above article in 1 sentences.

**Target**

The IT systems of three US hospitals have been infected with ransomware, which encrypts vital files and demands money to unlock them.

**Llama 2 7B: Full**

Hospitals in California and Kentucky have been hit by a ransomware attack.

**Llama 2 7B : Adaptive Wanda (50%)**

Three hospitals in the US have been hit by ransomware.

**Llama 2 7B: GRIFFIN (50%)**

Three US hospitals were hit by ransomware, which has affected their computer systems.

**Gemma 7B: Full**

A computer virus has hit three hospitals in the US, causing them to shut down their systems and activate back-up servers.

**Gemma 7B: Adaptive Wanda (50%)**

Three hospitals in the US have been hit by ransomware attacks.

**Gemma 7B: GRIFFIN (50%)**

Hospitals in the US have been hit by a ransomware attack, which has encrypted their files and demanded a ransom.

Figure 8: Generation examples by Llama 2 7B and Gemma 7B, given a prompt (BBC, 2016) from XSum (1-shot).

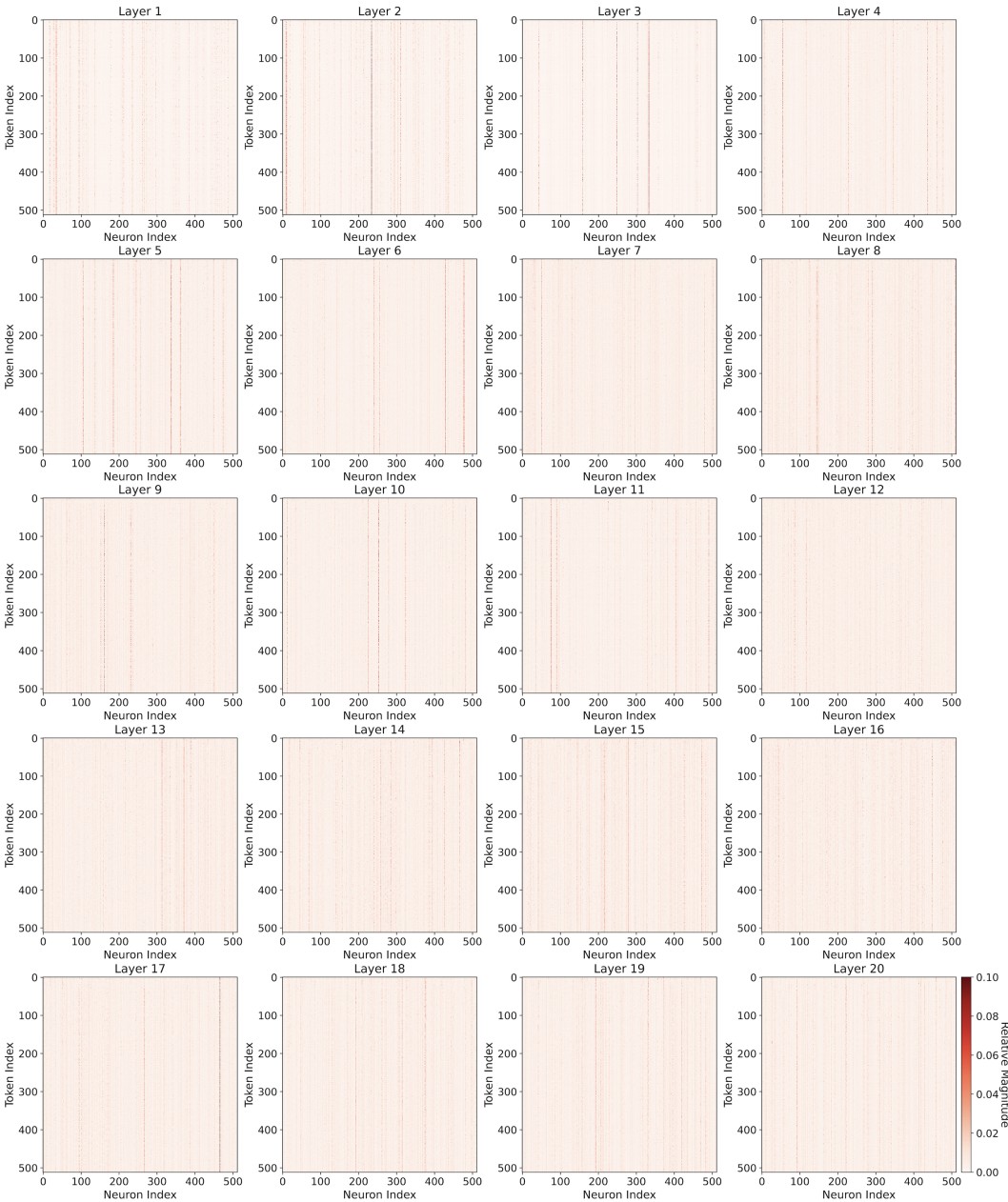

Figure 9: First 512 tokens and their relative FF activation magnitudes in layers 1 to 20 of Gemma 7B when inputting a sequence from PG-19.

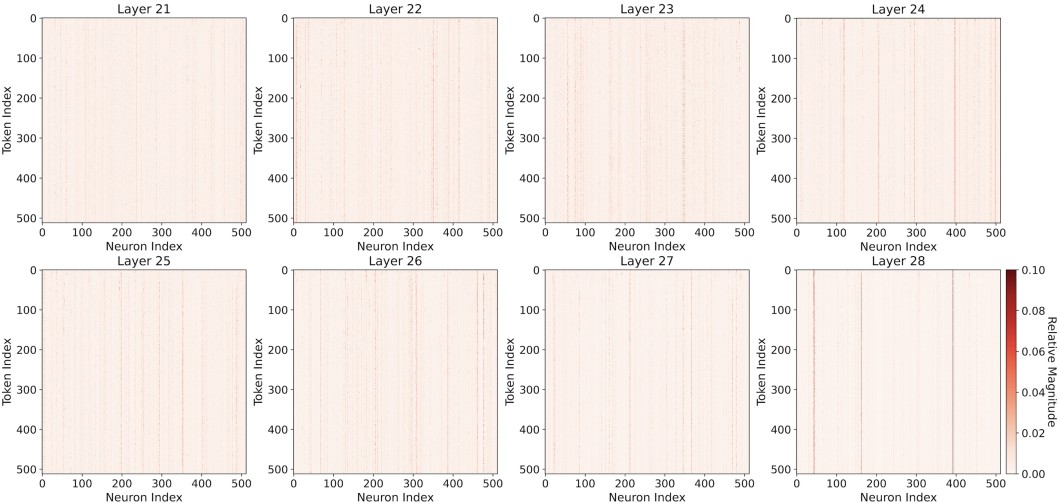

Figure 10: First 512 tokens and their relative FF activation magnitudes in layers 21 to 28 of Gemma 7B when inputting a sequence from PG-19.

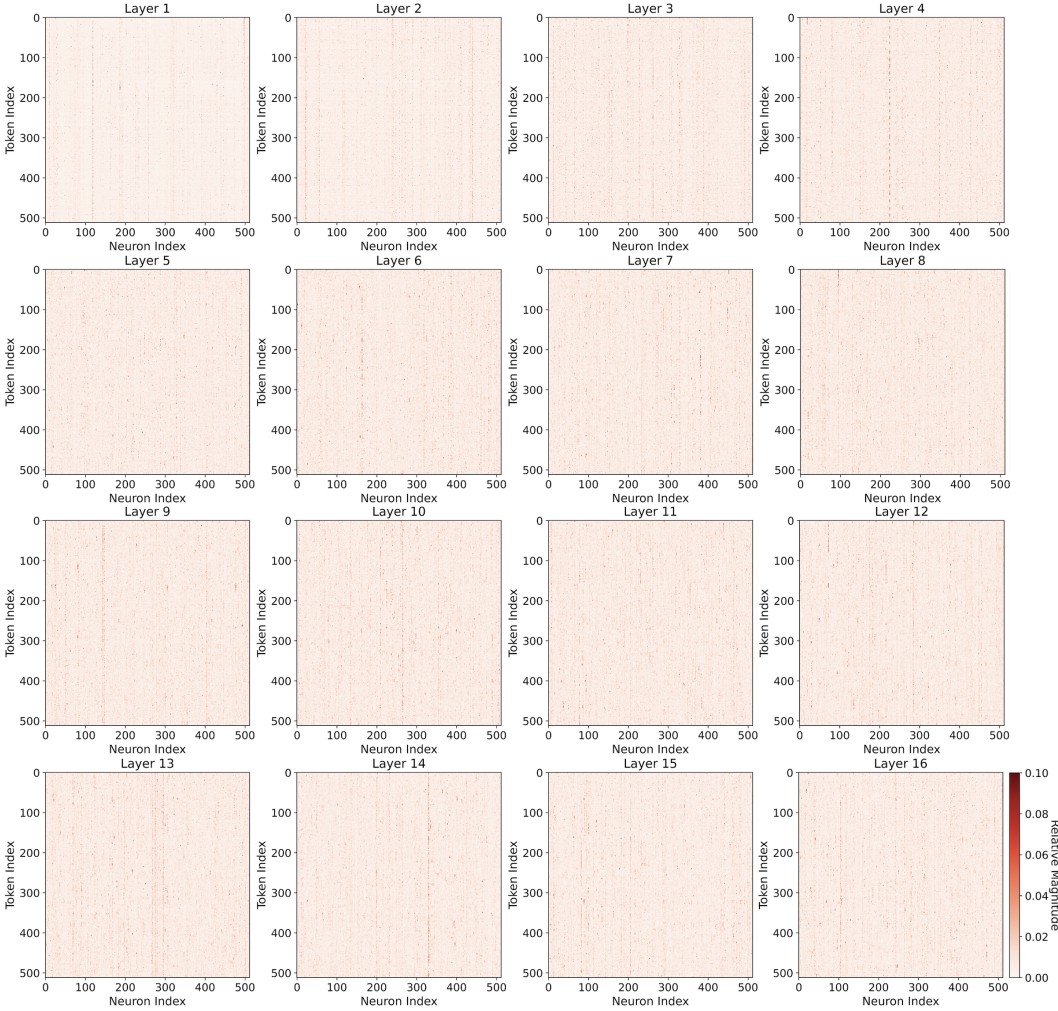

Figure 11: First 512 tokens and their relative FF activation magnitudes in layers 1 to 16 of Llama 2 7B when inputting a sequence from PG-19.

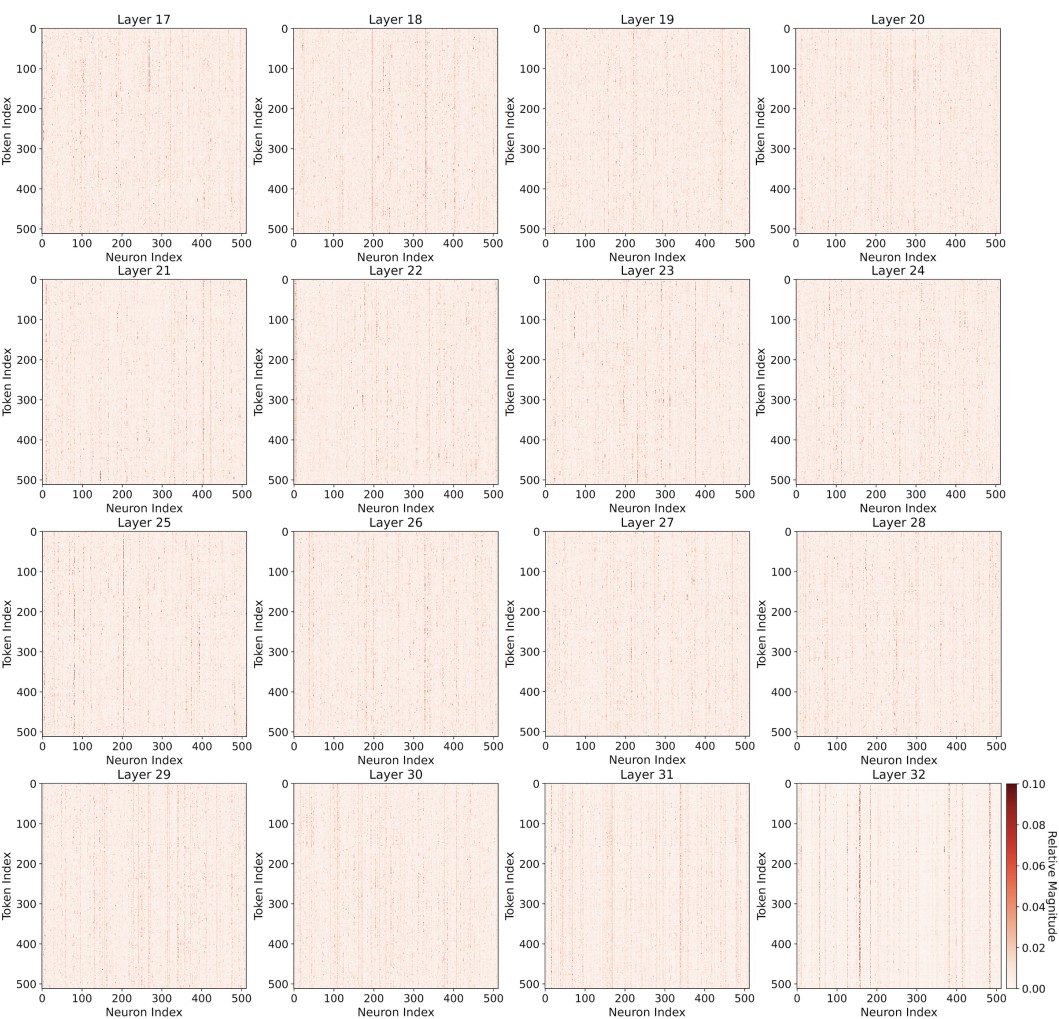

Figure 12: First 512 tokens and their relative FF activation magnitudes in layers 17 to 32 of Llama 2 7B when inputting a sequence from PG-19.

