# OpenReview forum: "Prompt-prompted Adaptive Structured Pruning for Efficient LLM Generation"
_colmweb.org/COLM/2024/Conference — COLM_

### Official Review · Reviewer_6mFc · 2024-05-02

**Rating:** 6
**Confidence:** 4
**Ethics Flag:** 1

**Summary:**

This paper introduces an approach for dynamically pruning the FF of Transformer LLMs that doesn't require any extra training. The structured pruning is conditioned on the prompt. When computing the prompt the regular full model is used. Then, based on the prompt activations, only the top-k rows of each FF across all layers are being kept. The generation is performed with this pruned network. So essentially the  pruning is dynamic at the sequence level. The authors present motivation for this approach by describing a behavior they call "flocking" where the activated neurons are mostly similar within a sequence but vary across different sequnces (studied on open source 7B and 13B models). They show that they can prune around 50% of FF for generation and mostly preserve classification and generation quality leading to about 1.25X speedup in LLama 13B.

**Questions To Authors:**

1. Can you define exactly what you mean by relative FF activation?
2. It seems by the heatmaps that maybe the distribution is slightly different across layers. Have you tried using different k across layers?

**Reasons To Accept:**

1. Simple but effective approach that doesn't require any extra training
2. Approach seem to work on several different LLMs and datasets
3. doesn't require any training so should be very easy to try on any open model

**Reasons To Reject:**

1. The writing and framing needs a bit improvement. The title says MoE but then the paper goes back and forth between MoE and pruning. The actual method is doing dynamic structured pruning. I don't think MoEs should be mentioned anywhere (maybe other than related work) and certainly not in the title since its confusing the reader.
2. Given that the method is actually pruning, I think the main weakness of the paper is lack of comparison to more pruning methods other than the presented naive magnitude approach.
3. The ablations part can also be improved. I am not sure what is the point of the sampling-based selection experiment. Instead, it will be interesting to see more ablations on the dynamic requirement for example if you take top-k when averaging activations over the full dataset and then keeping it static.
4. The method seems to work well on short generations but on longer ones the performance seems to substantially degrade. The authors suggest to use a long prompt for alleviating this but this defeats the efficiency purpose.
5. There are some more limitations to this method like having to select columns or rows for structural pruning rather than selecting from all neurons, and overall the speedup potential is limited with this pruning as even 50% parameter reduction of FF leads to small speedup as observed in experiments.

---

> ### Author Rebuttal · Authors · 2024-05-30
>
> Thank you for the detailed feedback! We address each concern/question below.
>
> Concerns:
> 1) Thanks for pointing this out, we can reframe the paper as a dynamic structured pruning method in our next version.
> 2) We designed a new baseline inspired by Wanda [2], a popular pruning method. The full model is used for the prompt while FF weights selected by Wanda based on the input’s features are used during generation (Adaptive Wanda). Comparisons between GRIFFIN and Adaptive Wanda are found in Table R1 under Reviewer cGRh. Both perform similar to the full model, but GRIFFIN has the added benefit of highly structured pruning whereas Wanda is unstructured.
> 3) Sampling-based methods show greedy top-k is best and are additional baselines. Fixing neurons across a dataset is interesting! In Table R3, we test two ways to select fixed FF neurons for generation which we will add to our next version. GRIFFIN is more performative in most cases, implying a need for adaptation.
>
> Table R3: 1-shot XSum and CNN/DailyMail Rouge-1 with fixed expert neurons at 50% sparsity. Shot Top-k selects FF experts based on $\mathbf{s}$ in the example shot. Global Top-k selects FF experts based on the average $\mathbf{s}$ across a dataset’s prompts.
> |Model|Method|XSum|CNN/DM|
> |-|-|-|-|
> |Llama 2 7B|Full|27.15|10.08
> ||Shot Top-k|21.11|10.30
> ||Global Top-k|23.28|10.84
> ||GRIFFIN|24.75|10.97
> |Gemma 7B|Full|26.86|17.45
> ||Shot Top-k|19.76|15.59
> ||Global Top-k|22.48|20.96
> ||GRIFFIN|25.86|18.26
>
> 4) A longer prompt is unlikely to harm efficiency much: Table 3 shows latency is dominated by generation, e.g. Llama 2 7B spends ~8.5% of the total latency on prompts.
> 5) GRIFFIN applies to FF blocks because it uniquely has flocking. This also makes it versatile as other efficient techniques can be independently deployed in attention (e.g. KV compression) and embedding layers that could further accelerate inference.
>
> Questions:
> 1) Relative FF activations are FF activations scaled down to unit vectors. If z is the vector of FF activations of a token, relative FF activations of that token is z / ||z||_2.
> 2) We noticed this too and initially played around with some simple ways to vary k across layers (e.g. skipping some layers and linear scheduling), but we did not notice much effect on performance. In future work, it would be interesting to explore more ways to choose k, for different layers and different samples.
>
> [2] Sun, M. et al, A simple and effective pruning approach for large language models, 2023.

---

### Official Review · Reviewer_WEew · 2024-05-08

**Rating:** 6
**Confidence:** 4
**Ethics Flag:** 1

**Summary:**

The article introduces a new, training-free Mixture of Experts (MoE) method called GRIFFIN. This method capitalizes on a phenomenon observed in large language models (LLMs) known as "flocking," where specific feed-forward (FF) neurons exhibit consistently higher activation relative to others within sequences. GRIFFIN leverages these observations to selectively activate the most relevant FF neurons for efficient processing, thereby maintaining model performance while reducing computational demands.

**Reasons To Accept:**

- GRIFFIN improves computational efficiency by reducing the number of active neurons during the model's forward pass, which lowers memory usage and speeds up processing without extensive hardware requirements.

- Despite reducing the number of active parameters in FFN by up to 50%, GRIFFIN maintains near-original performance levels across multiple tasks, which is a significant achievement compared to other sparsity-inducing methods.

- Unlike traditional MoEs that require retraining or fine-tuning, GRIFFIN operates without any additional training, making it easy to implement with pre-trained models.

**Reasons To Reject:**

Scalability Limitation Due to Batch-Size Constraint: GRIFFIN's current implementation is constrained to a batch size of one, which limits the model's ability to leverage parallel processing capabilities inherent in modern computational hardware. While a batch size of one achieves a modest speed-up of approximately 1.2x, this benefit does not scale linearly with larger batch sizes. This is due to the unique 'flocking' patterns observed in individual samples, which may vary significantly across different inputs within a batch. This variability makes it challenging to maintain consistent efficiency gains when processing multiple samples simultaneously. Such constraints on batch size could restrict the method's utility in real world model severing.

---

> ### Author Rebuttal · Authors · 2024-05-30
>
> Thank you for the detailed review of our work, especially on real world considerations for deployment!
>
> Based on your suggestions, we performed experiments on batching with highly promising results. Within a batch, we sum the selection statistic $\mathbf s_i$ of each sample, scaled inversely by each sample’s root  prompt length, $S_i$, i.e. $\mathbf{s} = \sum_{i} \mathbf{s}_i \sqrt{S_i}$. Then, the same experts for all samples within a batch would be selected based on $\mathbf{s}$. Results in Table R2 convey that GRIFFIN is also robust to batching, as performance degrades very slowly as we increase the batch size. As such, while GRIFFIN was especially designed for single sample use cases (e.g. personal devices), our method scales impressively well to large batch sizes.
>
> Table R2: 1-shot XSum Rouge-1 scores for increasing batch sizes. For each batch, the same 50% of FF expert neurons are selected.
>
> |Batch Size|1|2|4|8|16|32|64|
> |-|-|-|-|-|-|-|-|
> |Llama 2 7B|24.75|24.05|24.26|24.04|23.75|23.61|23.49|
> |Gemma 7B|25.86|25.55|25.01|24.20|24.37|23.32|23.12|
>
> In the next version of our paper, we will include the results in Table R2 in the form of a graph. Let us know if you have any further questions and comments.

---

> > ### Comment · Reviewer_WEew · 2024-06-02
> > **Thank you for the rebuttal**
> >
> > Thank for very much for the new result, and it looks quite promising. I will raise the score.

---

### Official Review · Reviewer_yRHP · 2024-05-09

**Rating:** 7
**Confidence:** 3
**Ethics Flag:** 1

**Summary:**

This paper discusses the phenomena of “flocking”, where FF activations in dense LLMs tend to concentrate by column within the same sequence, and proposes GRIFFIN, a novel technique that selects subsets of FFN parameters per sample based on flocking pattern on the prompt to achieve inference speed up without much loss of quality. The proposed method is training-free, empirically applicable to a list of LLMs, and simple to implement.

**Questions To Authors:**

1. By end of page 1 - “Perhaps an even bigger weakness of many of these methods is the inability to effectively carry over to pre-trained LLMs with non-ReLU activations.” - this is not evident to me, could you elaborate, or if this is from previous studies, cite the study?
2. Figure 3 - Rows of Z_bar are not unit vectors. Is this a mistake in the figure?
3. Table 2 - Llama 2 13B is significantly worse than 7B in CNN-DM, and 2.5 pts of ROUGE-1 is pretty much no overlap. Mistral 7B also sees this issue. What is happening for these experiments? What do the models generate?
4. The observation that flocking patterns are shared within 1 sequence and differ across sequences begs one question - what happens when you take 2 sequences that have distinct patterns and concat them together? will it create problems for GRIFFIN? the meta-question here is what causes patterns to stay stable within sequence, and will there be naturally occurring data that breaks this assumption (therefore potentially make GRIFFIN fail on them)?
5. Section 4.2 right before eqn 6 - “...we do this by aggregating taking…” → "...we do this by aggregating..."?

**Reasons To Accept:**

The algorithm this paper proposes is well-motivated and empirically strong. GRIFFIN tackles the important problem of efficient inference, achieving a decent boost in latency without costly retraining, and is empirically demonstrated to be widely applicable.

This paper also highlights the flocking behavior and shows that it is possible to exploit such inherent structure for practical speed up. I can see this work spark future ideas along this topic.

**Reasons To Reject:**

I don’t find major reasons to reject the paper. Some lesser issues I have with the paper are

1. There is not much quantitative characterization of the flocking behavior. The paper shows many examples of flocking, but it’d be good to have some kind of quantitative measurement of this behavior at scale, to demonstrate how ubiquitous is the flocking behavior "in the wild".
2. ROUGE scores for Llama-13B and Mistral-7B on CNN-DM seems too low without any explanation (also Llama-13B < Llama 7B). This brings a bit doubt to the experiment results. It’d be good to provide some qualitative generation examples.
3. In tandem to #2 - it’d be great to provide generation samples with GRIFFIN, as ROUGE-based evals are n-gram-overlap based and may hide problems. It’d be good to rule out possibilities such as generating incoherent yet overlapping phrases, etc., by showing some qualitative samples.
4. A pedantic point: to posit GRIFFIN as an approach to convert a dense model to MoE seems a bit of a stretch of concept. GRIFFIN seems to have much more affinity to adaptive pruning or adaptive (structured) sparsity rather than MoE.

---

> ### Author Rebuttal · Authors · 2024-05-30
>
> Thank you for reviewing our submission and highlighting some key takeaways! We address your concerns/questions below. Please let us know of any lingering questions and feedback.
>
> Concerns:
> 1) We can include similarity plots between tokens’ relative activations. Since we are unable to share images for the rebuttal, we roughly describe them. Using the same sample in Figure 1, many Gemma 7B layers had high cosine similarity (0.6 to 0.8) between tokens’ activations. Most of Llama 2 7B’s hovered around 0.35 to 0.4.
> 2) We can add example generation outputs. For 1-shot CNN/DM, Mistral 7B and Llama 2 13B often produced a sequence of “\n” or repeated items from the example shot. With 3-shot, this is not an issue (>25 Rouge-1 for both).
> 3) We will also include some example generation outputs from GRIFFIN and the full model in the next version.
> 4) Thanks for bringing this to our attention. We will reframe our work as dynamic structured pruning in our next paper version.
>
> Questions:
> 1) We can elaborate on this in the next paper version. For example, the MoEfication algorithm [3] fine tunes non-ReLU activations to ReLUs before constructing MoEs, but it is not the only one to require exact sparsity in FF activations [4, 5]. In turn, these have motivated fine tuning LLMs to have ReLU activations [6, 7].
> 2) Row vectors in Figure 3 are approximately l2 unit vectors. For illustration purposes, we rounded to one decimal place, inducing slight rounding errors.
> 3) See #2 in “Concerns”.
> 4) In our evaluations with XSum and CNN/DailyMail, the example shot and test samples are very different, yet GRIFFIN still appears to do well. The cause of flocking is of great interest to us. Flocking occurs in many different tasks–Figure 6 shows that it even occurs with nonsense inputs. Greater analysis in flocking in terms of cause and interpretability is something we are excited to work on in the future!
> 5) Thank you for catching this typo! We will correct it in the next version.
>
> [3] Zhang, Z., et al., Moefication: Transformer feed-forward layers are mixtures of experts, 2021.
>
> [4] Csordás, R., Irie, K., & Schmidhuber, J., Approximating two-layer feedforward networks for efficient transformers, 2023.
>
> [5] Li, Z. et al., The lazy neuron phenomenon: On emergence of activation sparsity in transformers, 2022.
>
> [6] Mirzadeh, I. et al., Relu strikes back: Exploiting activation sparsity in large language models, 2023.
>
> [7] SpaseLLM Team, Sparse large language models with relu activation, 2023.

---

### Official Review · Reviewer_cGRh · 2024-05-13

**Rating:** 5
**Confidence:** 4
**Ethics Flag:** 1

**Summary:**

This paper introduces GRIFFIN, an interesting MOE method, which leverages special activation patterns in Feedforward layers for pruning. The key feature of GRIFFIN lies in its training-free design for selecting unique FF experts at the sequence levels. Besides,  this method is adaptable to non-ReLU methods, which makes it practical in applications, given that many popular LLMs use other activations like SwiGLU. Experimental results demonstrate that, even with only 50% of FF parameters, an MOE LLM employing GRIFFIN can still maintain performance comparable to dense models.

**Questions To Authors:**

Please refer to the "Reasons to Reject"

**Reasons To Accept:**

1. The proposed method stands out for its training-free design, a practical advantage particularly beneficial for large-scale LLMs. The discovery of fixed patterns within Feedforward Neural Networks (FFNs) is an intriguing observation, evident across various LLM architectures such as LLaMA and Gemma, with activations like SwiGLU and GEGLU. This consistency underscores the potential compressibility of FFN layers.

2. GRINFFIN's simplicity is notable, as it relies solely on the L2-norm of activations during runtime. This streamlined approach not only enhances computational efficiency but also simplifies the implementation. Moreover, the selected experts, i.e., the pruned layers identified by GRINFFIN, can be seamlessly integrated into the decoding process for generating future tokens.

**Reasons To Reject:**

The main limitation of this paper is the absence of baseline methods in the experimental section. For instance, Dejavu[1] introduces a contextual structured sparsity approach to dynamically prune layers during runtime. This parallels the strategy adopted in this paper, which also focuses on adaptively pruning Feedforward Neural Networks (FFNs) for different inputs. However, the experiments presented in this submission solely involve magnitude pruning, which may not offer a robust baseline compared to more sophisticated methods like Dejavu. Integrating diverse baseline approaches could provide a more comprehensive assessment of the proposed method's performance and elucidate its relative strengths and weaknesses in comparison to existing techniques.

---

> ### Author Rebuttal · Authors · 2024-05-30
>
> Thank you for your valuable comments and curiosity in our baselines! We address your comments below. Please let us know if you have any follow-up questions.
>
> As a new baseline, we use Wanda [2] during generation. This baseline is stronger than the magnitude selection method and is well-known in the pruning community. In more detail, the full model is still used during the prompt phase while FF weights selected by Wanda using the current inputs’ features are used during generation (Adaptive Wanda). Comparisons between GRIFFIN and Adaptive Wanda can be found in Table R1.
>
> Table R1: Adaptive Wanda and GRIFFIN on XSum (1-shot), CNN/DailyMail (1-shot), CoQA (0-shot), SCROLLS QASPER (0-shot) at 50% FF sparsity.
>
> |Model|Method|XSum (Rouge-1/2/L)|CNN/DM (Rouge-1/2/L)|CoQA (F1/EM)|QASPER (F1)|
> |-|-|-|-|-|-|
> |Llama 2 7B|Full|27.15/9.06/22.62|10.08/0.13/9.55|77.35/63.88|26.31|
> ||Adaptive Wanda|**25.59/8.18/21.34**|9.90/0.26/9.39|77.16/63.53|**27.61**|
> ||GRIFFIN|24.75/7.41/20.55|**10.97/0.66/10.37**|**77.18/63.58**|25.76|
> Gemma 7B|Full|26.86/9.15/22.03|17.45/4.15/15.94|79.04/65.25|30.78|
> ||Adaptive Wanda|24.76/7.79/20.39|12.10/2.21/11.20|75.79/61.87|**30.62**|
> ||GRIFFIN|**25.86/7.81/20.93**|**18.26/4.75/16.58**|**78.52/64.62**|27.37|
>
> We observe that GRIFFIN and Adaptive Wanda are both great at preserving the original model’s performance. However, we note that Wanda’s pruning algorithm is unstructured, meaning it cannot be easily translated to real latency improvements. Meanwhile, GRIFFIN is able to be performative despite the structural constraint we imposed. We will add this baseline (including its performance with other models) in our next paper version.
>
> Following up on your comments on Deja Vu [1], we recognize its ability to dynamically select neurons during inference, but we do not think it is comparable. A major reason is that Deja Vu requires data curation, training, and extra parameters, so compared to GRIFFIN (which requires no preparation), there is a significant gap in resource requirements. Unlike Deja Vu, our new baseline, Adaptive Wanda, selects FF neurons depending on the input without training or data curation and exhibits competitive performance.
>
> [1] Liu, Z. et al, Deja vu: Contextual sparsity for efficient llms at inference time, 2023.
>
> [2] Sun, M., Liu, Z., Bair, A., & Kolter, J. Z, A simple and effective pruning approach for large language models, 2023.

---

### Author Response · Authors · 2024-06-07
**Thank you to the reviewers!**

We thank all the reviewers again for making thorough comments and suggestions! Since the discussion period will come to a close soon, if  not done so already, we would greatly appreciate if the reviewers could let us know if we have addressed their remarks, so we can further improve the quality of our paper. Thank you!

---

### Decision · Program_Chairs · 2024-07-10

**Decision:**

Accept

**Comment:**

Ngl this is an interesting paper. but the framing of this paper is quite haphazard.

Reviewers generally like this paper and recommend acceptance. I am leaning accept too. However, i really think the framing needs to be less confusing and more clear. This is a fundamentally weight sparsity and pruning paper and i think the connection with MoE can be a little distracting. I think this paper would benefit from a more direct framing about what it actually is. Conceptually, drawing in MoE in the narrative makes the paper hard to parse.

I would in fact start the paper by explaining and coining the term flocking and explaining why griffin leverages it. There are too many random plot points happening in this paper.

In short, i recommend acceptance but authors could benefit from cleaning up the story a bit. this paper is trying too hard. a more down to earth narrative that is simple will likely make more impact.

[comments from the PCs] We strongly recommend following the AC advice to improve the paper.